# The Formation Mechanism of Nanocrystals after Martensitic Transformation

**DOI:** 10.3390/ma15186258

**Published:** 2022-09-08

**Authors:** Song-Jie Li, Shi-Long Su, Guan-Jie Hu, Qiang Zhao, Zheng-Yu Wei, Yun Tian, Cheng-Duo Wang, Xing Lu, De-Hai Ping

**Affiliations:** 1School of Chemical Engineering, Zhengzhou University, 100 Science Avenue, Zhengzhou 450001, China; 2School of Materials Science and Engineering, Zhengzhou University, 100 Science Avenue, Zhengzhou 450001, China; 3School of Materials Science and Engineering, Dalian Jiaotong University, Dalian 116026, China

**Keywords:** carbon steels, martensite, substructure, nanocrystal, transmission electron microscopy (TEM)

## Abstract

Understanding the ultrafine substructure in freshly formed Fe-C martensite is the key point to reveal the real martensitic transformation mechanism. As-quenched martensite, whose transformation temperature is close to room temperature, has been investigated in detail by means of transmission electron microscopy (TEM) in this study. The observation results revealed that the freshly formed martensite after quenching is actually composed of ultrafine crystallites with a grain size of 1–2 nm. The present observation result matches well with the suggestion based on X-ray studies carried out one hundred years ago. Such nanocrystals are distributed throughout the entire martensite. The whole martensite shows a uniform contrast under both bright and dark field observation modes, irrespective of what observation directions are chosen. No defect contrast can be observed inside each nanocrystal. However, a body-centered cubic {112}<111>-type twinning relationship exists among the ultrafine α-Fe grains. Such ultrafine α-Fe grains or crystallites are the root cause of the fine microstructure formed in martensitic steels and high hardness after martensitic transformation. The formation mechanism of the ultrafine α-Fe grains in the freshly formed martensite will be discussed based on a new γ → α phase transformation mechanism.

## 1. Introduction

In metals and alloys, the hardness normally increases with the decreasing grain size. In steels, particularly carbon steels, a well-known phase transition (martensite transformation) occurs when the steel is cooled down quickly from the austenitization temperature. This transformation process commonly has a pronounced effect on the mechanical properties, particularly hardness, since a fine microstructure forms in the steel after this transformation. As the mechanical properties are significantly affected by this transformation, the martensitic transformation mechanism has been an attractive topic in the fundamental research field of steels. Nevertheless, the ultimate fine α-Fe produced via this transformation process remains unclear to date. In order to observe the fine α-Fe precisely, it is necessary to conduct an in-depth characterization of the substructure in the freshly formed martensite right after martensitic transformation.

In Fe-C binary alloys, the martensitic transformation begins at the martensite start temperature (Ms), which decreases as the carbon content increases. In general, the microstructural investigations on the as-quenched martensite are carried out at room temperature. Thus, the observed martensitic structure at room temperature has unavoidably experienced an auto-tempering process at the temperature interval between Ms and room temperature. In the Fe-C binary alloy system, the martensite, which begins to transform at a temperature close to room temperature, can then be treated as a freshly formed martensite since its substructure experiences little auto-tempering. Even if it is kept at room temperature, the as-quenched martensite unavoidably suffers the auto-tempering treatment or the natural aging upon exposure to air. Thus, it is better to investigate the as-quenched martensite as soon as quenching treatment is completed. The martensite structure was initially investigated by using optical microscopy and X-ray diffraction. The initial X-ray studies on the crystal structure of iron and steel has suggested that the martensite is composed of a 2-nm fine structure [1]. Later, the as-quenched high carbon martensite was regarded as a single crystal [2,3,4,5,6,7,8,9,10,11]. Nevertheless, unlike large carbide particles, large martensite plates have never been successfully extracted from the as-quenched samples for X-ray characterization. It is also impossible to characterize the martensite substructure by an optical microscope since the resolution is too low. Later in 1960s, a high density of body-centered cubic (bcc) {112}<111>-type twins were observed in the as-quenched high carbon martensite [12,13,14,15,16,17,18,19,20]. Obviously, the existence of the twins has indicated that the high carbon martensite is no longer a single crystal due to the presence of twinning boundaries.

Recent results carried by TEM observations on the as-quenched martensite have revealed that nanoclusters of α-Fe formed in the as-quenched high-carbon martensite [21,22]. However, contrast comparison with retained austenite has not been studied, and the formation mechanism of such ultrafine α-Fe grains has not been explained yet. The results about the observations of the ultrafine α-Fe crystals in as-quenched martensite are quite surprising since it is contradictory to the traditional description of the Fe-C martensite structure. Thus, the martensite substructure of cast-iron alloys will be investigated in detail by means of TEM in this study. The cast-iron alloy is a high-carbon alloy with a comparative volume fraction of retained austenite and quenched martensite structure in the as-quenched state. Thus, it is easier to use the TEM technique to observe both phases in a small observation field.

## 2. Experimental

In carbon steels, the Ms temperature decreases as carbon content increases during quenching. In order to drop Ms down to around room temperature, the carbon content required in binary Fe-C alloys is normally about 1.8 wt.% or higher [23]. A cast-iron alloy (chemical composition: C: 3.4, Si: 1.8, Mn: 0.6, S: 0.08, P: 0.15 (wt.%)) was selected for the present investigations. A cast iron plate with the size of 20 mm × 20 mm × 2 mm was cut from bulk alloy. The plate was heat treated at 1150 °C for 1 h, followed by water quenching. The TEM specimen was extracted from the water-quenched plate, and was then subject to mechanical grinding and polishing, and finally ion-milling at room temperature. The microstructural observation was carried out with a TEM (FEI Talos F200s TEM was made in Czech) operated at 200 kV and a scanning electron microscope (SEM) (FEI Nano450 SEM was made in Czech) operated at 20 kV.

## 3. Results

Figure 1 shows the morphology and twin density of the martensite structure in as-quenched cast iron samples. A plate-like or lenticular martensite morphology is observed, as shown in Figure 1a. The two martensite plates in Figure 1a are both off zone axes, which results in the bright contrast. After one of the martensite plates was tilted to align with the zone axis, which means the zone axis is parallel to the incident electron beam, the martensite shows black contrast under a bright field mode, as shown in Figure 1b, and it is difficult to observe a detailed substructure inside the martensite. However, the corresponding dark field image (Figure 1c) clearly reveals a high density of twins. In an enlarged dark field image (Figure 1d), it can be clearly seen that the twin width is several nanometers, and the thinner ones are just 1–2 nm. In the twinned part (bright-contrast region), the smallest α-Fe crystal is also 1–2 nm. In Figure 1d, if the bright contrast is treated as the twinned crystal part, then the dark contrast region corresponds to the matrix, and the ω-Fe phase particles are distributed in the twinning boundary region [24]. The ultrafine α-Fe grains in the matrix region and those in the twinned region have a BCC {112}<111>-type twinning relationship rather than a twinned crystal with a sharp twinning boundary to split one crystal into two parts (matrix and twinned parts). This is due to the existence of the ω-Fe phase distributed in the twinning boundary.

Figure 2 shows a general TEM observation for an as-quenched cast iron sample. The images in Figure 2a,b were obtained in a TEM bright field and dark field modes, respectively. In general, compared to the bright field image, the TEM dark field image can show a better or higher contrast for observing a much finer structure. In Figure 2a, a region containing fringes, which are caused by the sample thickness variation, is outlined by the red dashed circle. Except for the contrast of the thickness of the fringe, the whole martensite in both the bright and dark field images shows a uniform contrast. Such a uniform contrast can be easily found in Figure 3, which is another dark field image showing a very fine structure. The whole martensite structure shows a uniform contrast no matter how large it is. However, the contrast is not uniform across the austenite, and the dislocation-like contrast can be clearly observed.

The TEM image contrast can be caused by several reasons, such as the second phase contrast, grain orientations, thickness contrast, defects (dislocations, twin boundaries, etc.), element segregation contrast or chemical composition variation contrast, and so on. It is quite difficult to explain the origin of the image contrast in a TEM sample. However, the present TEM sample consists mainly of iron, with the light element carbon more or less uniformly distributed in the austenite. The contrast variation inside one austenite grain can be caused by the difference in the local thickness in the specimen and/or defects (sub-grain boundaries or dislocations). 

Figure 3 shows a dark field TEM image containing both an austenite and martensite structure. The austenite region can be simply recognized from the contrast difference between the two structures (martensite and austenite). In the enlarged image on which a local martensite region has been inserted, neither a dislocation contrast nor an obvious contrast caused by the thickness variation can be seen. From the enlarged image, it is easy to see the contrast from ultra-fine grains with almost the same crystal orientation and the grain size of 1–2 nm. Such fine grains are the only reason for the formation of the uniform contrast in martensite. Thus, in an as-quenched high-carbon martensite, the fresh martensite is composed of ultra-fine grains and sub-grain boundaries between them. Since Fe is the main element in the cast iron alloy, and selected area electron diffraction (SAED) has confirmed that the nanocrystals have a body-centered cubic (bcc) crystal structure, these nanocrystals can be deemed as α-Fe grains with almost the same orientation. It can be estimated from the dark field images that the size of these nanocrystals is about 1–2 nm. Some regions in dark field images show a larger particle size, which is likely to be the aggregation of several ultrafine α-Fe grains due to very close orientation.

The martensitic substructure always shows a uniform contrast regardless of its orientation. Figure 4 shows the TEM observation with the electron beam parallel to the [$\bar{1}$13] axis of α-Fe. Both the bright field image (Figure 4a) and the corresponding dark field image (Figure 4c) show a clear contrast between the ultrafine particles and the surrounding matrix. The SAED pattern in Figure 4b has confirmed that two crystalline phases (α-Fe, ω-Fe(C)) coexist [24,25,26,27,28,29]. The dark field image was taken by using α-Fe (110) diffraction spot. Thus, both phases are nanocrystals with almost the same size.

The absence of the twinning contrast in the martensite is ascribed to the fact that the twinning planes are not parallel or close to the electron beam direction. When the twinning planes form a certain angle with the incident electron beam direction (the observation direction), the twin boundary ω-Fe_3_C ultrafine particles completely overlap with the α-Fe grains [24,25,26,27,28,29]. Thus, it is difficult to recognize the ω-Fe_3_C particles from the high-resolution TEM lattice image (Figure 4d) due to the lattice coherence between the two phases. From both the SAED pattern and the high-resolution lattice image, the α-Fe (110) plane completely overlaps with the ω-Fe_3_C (0$\bar{1}$1$\bar{1}$) plane, as indicated by the red dashed line in Figure 4d. The white dashed line corresponds to the ω-Fe_3_C (10$\bar{1}$0) plane and the interplanar spacing is exactly three times that of the α-Fe {112} planes, although it is unable to observe in the high-resolution lattice image due to the resolution limitation. 

The above experimental observations from different observation directions have revealed that the martensite substructure is composed of ultrafine α-Fe and ω-Fe_3_C grains in the as-quenched state regardless of the observation directions, and the grain size is about 1–2 nm. Observations from all directions show almost the same grain size, and 1–2 nm is the smallest size for crystalline phases in both metals and alloys. The formation mechanism about such a small size grain will be discussed later.

Figure 5 shows TEM observation from location on the as-quenched martensite sample. The bright field image and its corresponding SAED pattern are shown in Figure 5a,b, respectively. The SAED pattern is similar to that shown in Figure 4b so that the index is not shown in Figure 5b in order to exhibit the diffraction spots more clearly. The dark field images shown in Figure 5c–f were obtained by using different diffraction spots, as indicated by the yellow arrows. In all of these dark field images, nanocrystals with sizes of 1–2 nm were observed inside the martensite. As can be seen from one of the enlarged images in Figure 6, the fundamental structure inside the martensite is composed of nanocrystals alone, and the size of all these nanocrystals is about 1–2 nm.

The TEM dark field images have shown that the martensite substructure is composed of nanocrystals no matter what observation directions and diffraction spots were used. Since the grain size is about 1–2 nm, which is the smallest crystalline size for pure Fe crystal, the martensite in this condition can be treated as freshly formed martensite suffering few auto-tempering effects. In fact, all twinned martensite consists of nanocrystals as its substructure [21,22]. 

## 4. Discussion

The martensite structure is formed in the austenite (γ-Fe) matrix with an fcc crystal lattice structure via a well-known γ → α transformation in pure iron. In addition, the martensite consists of two crystalline phases, namely α-Fe and ω-Fe_3_C, in Fe-C binary alloys [24,25,27,28]. The detailed γ → α transformation pathway has been explained in the previous publication [30]. Here, emphasis will be placed on the formation mechanism of the α-Fe grains with a size of about 1–2 nm. The present observation result actually matches well with the suggestion based on X-ray studies carried out one hundred years ago [1]. At that time, by comparing the X-ray diffraction line width obtained from an extremely fine-grained gold colloid, the author (A. Westgren) has given a clear suggestion that the martensite is composed of a 2-nm-particle structure.

The schematic in Figure 7 shows the γ → α transformation pathway. Figure 7a is the model presented in ref. [30]. Figure 7b is the Bain model, which has been known and widely accepted for a long period [7,8,9,10]. Figure 7c is the projection of Figure 7a along the *c*_fcc-Fe_ direction. The red dashed lines correspond to the projection of α-Fe along its <110> direction, and the blue dashed line is parallel to both the (1$\bar{1}$0)_γ-Fe_ plane and one of the {112}_α-Fe_ planes. Those atoms on the blue dashed line are kept unchanged during the transformation, while the atoms in different layers in the shadow region, which are parallel to the blue dashed line, will experience a shift of either √5/20)*a*_fcc-Fe_ (~0.4014Å for *a*_fcc-Fe_ = 3.59 Å) or 2 × 0.4014 Å along the arrow direction, as shown in Figure 7c.

As explained in Figure 7d, the formation of a large three-dimensional α-Fe crystal requires a continuous shift of the atoms in other layers outside the shadow region. For example, the atoms at the “B” or “B′” positions in Figure 7d need to shift three times as long as 0.4014 Å (~1.204 Å). One more layer-shifting process results in an atom movement of about 4 × 0.4014 Å = 1.6056 Å, which is longer than half (~1.25 Å) of the distance of two neighboring atoms in that layer. Thus, one α-Fe grain can have a size of about 2 × a_fcc-Fe_ × $\sqrt{2}\;$(~10.15 Å and ~1 nm). This means that the α-Fe crystal cannot grow in size during the transformation process. This is believed to be the fundamental reason for the formation of 1–2 nm-size ultrafine α-Fe grains in the initially formed martensite structure. At the same time, the formation of α-Fe grains or the γ → α transformation can occur anywhere as long as there are no carbon atoms. In Figure 7d, the distance from “B” to “B′” is about 0.75 nm, which is still smaller than 1–2 nm estimated from the grain morphology in the TEM dark field images. Thus, a practical recrystallization among the nanocrystals must have occurred during the transformation. 

Based on the Bain model shown in Figure 7b, a large α-Fe grain could not form either. Actually, there is no suitable modeling to explain the formation of a large α-Fe crystal during the γ → α transformation process to date. Although, the α-Fe structure shown in Figure 7d can be formed without a size limit on the (1$\bar{1}$0)_γ-Fe_ plane based on the present atomic shift. In fact, numerous α-Fe unit cells will form at the same time everywhere. These homogeneously formed α-Fe unit cells on various (1$\bar{1}$0)_γ-Fe_ planes will possess almost the same <111>_α-Fe_ orientation on the (1$\bar{1}$0)_γ-Fe_ plane or one of the {112}_α-Fe_ planes, as experimentally revealed in Figure 4. A small mis-orientation can result in the formation of sub-grain boundaries between ultrafine α-Fe grains, and therefore, the nanocrystals form the initial martensitic substructure. Such a nanocrystal has been commonly observed in quenched twinned martensite regardless of the carbon content [21,22]. 

It is well-known that the martensitic transformation results in the formation of fine-grain α-Fe in steels. However, how small the fine α-Fe grain could be remains unclear. Based on the present experimental results, it is quite evident that the smallest size of the α-Fe grains formed after martensite transformation is just 1–2 nm. Upon tempering, α-Fe grains with different grain sizes can form via recrystallization [21]. Obviously, such tiny α-Fe grains cannot be formed by a long-range atomic shearing or displacive movement mechanism, which was proposed based on the single-crystal martensite with a relatively large crystal size. Such a nanocrystal in the martensite generates an enormous amount of sub-grain boundaries naturally, which results in a volume expansion of the martensite and thus the crack formation in the martensite. Figure 8 shows a SEM image revealing cracks in as-quenched martensite. These cracks, which were normally formed during the quenching process, pass through the whole martensite.

## 5. Conclusions

TEM dark field observations on the freshly formed martensite in as-quenched cast iron samples have revealed the following outcomes:(1)In the freshly formed martensite, the martensite substructure is composed of 1–2 nm nanocrystals with almost the same orientation.(2)Under TEM observations, martensite shows a uniform contrast, and no structural defect can be observed inside the grains due to the ultimate fine grain size.(3)The formation mechanism of the nanocrystals can be ascribed to the local collective movement of atoms.

## Figures and Tables

**Figure 1 materials-15-06258-f001:**
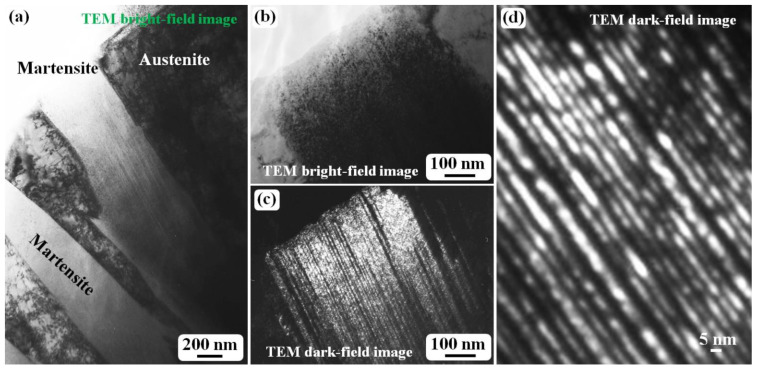
TEM images obtained from the water-quenched cast iron plates after austenitizing at 1150 °C for 1 h. (**a**) General bright field image revealing the martensite structure. (**b**) Bright field micrograph revealing twin contrast in martensite structure. (**c**) Dark field image revealing the twin size and morphology. (**d**) An enlarged dark field image revealing an ultra-high density of the twins.

**Figure 2 materials-15-06258-f002:**
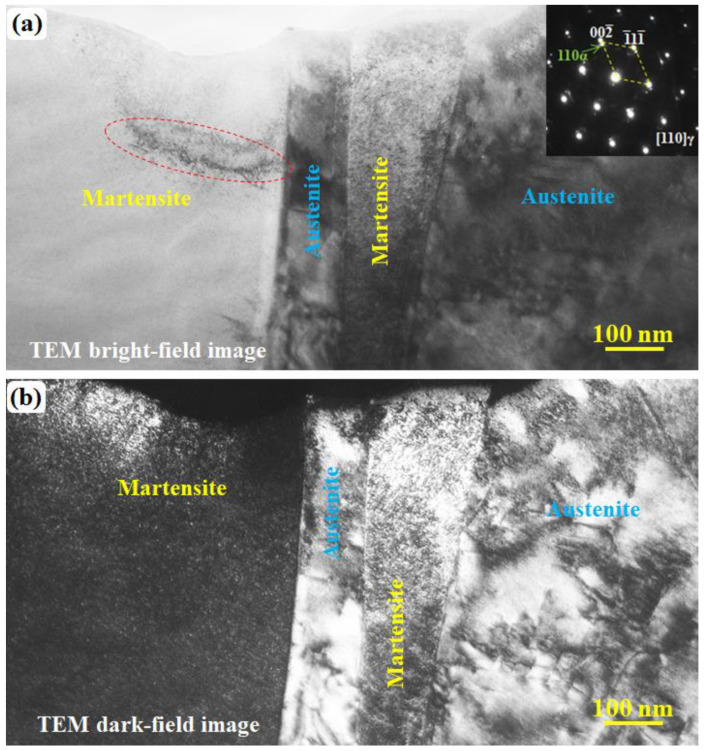
TEM bright field (**a**) and dark field (**b**) images revealing that the substructure in quenched martensite and retained austenite is different. The inset in (**a**) is the SAED pattern containing both austenite and martensite diffraction spots. The dark field image (**b**) was taken using the spots (00$\bar{2}$)γ and (110)α.

**Figure 3 materials-15-06258-f003:**
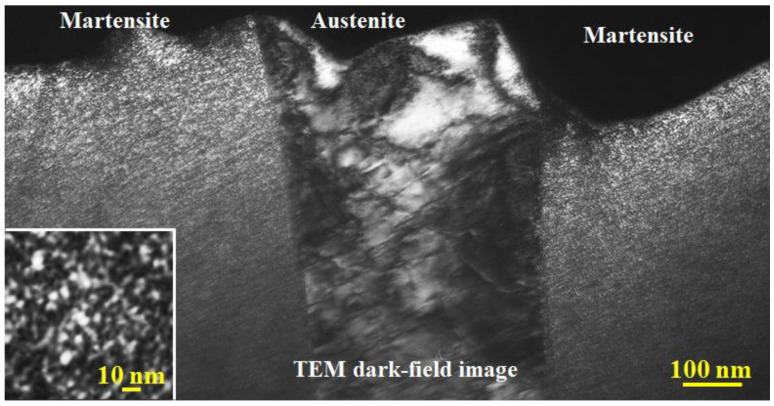
TEM dark field image revealing that the substructure in quenched martensite and retained austenite is different.

**Figure 4 materials-15-06258-f004:**
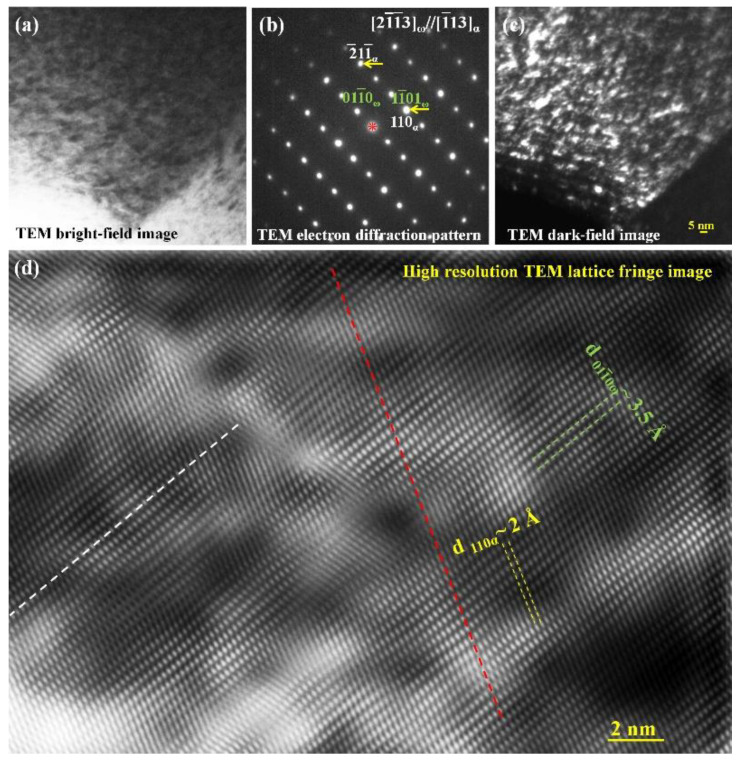
(**a**) TEM bright field image, (**b**) SAED pattern, (**c**) TEM dark field image, (**d**) high-resolution TEM image of as-quenched martensite in a cast iron sample.

**Figure 5 materials-15-06258-f005:**
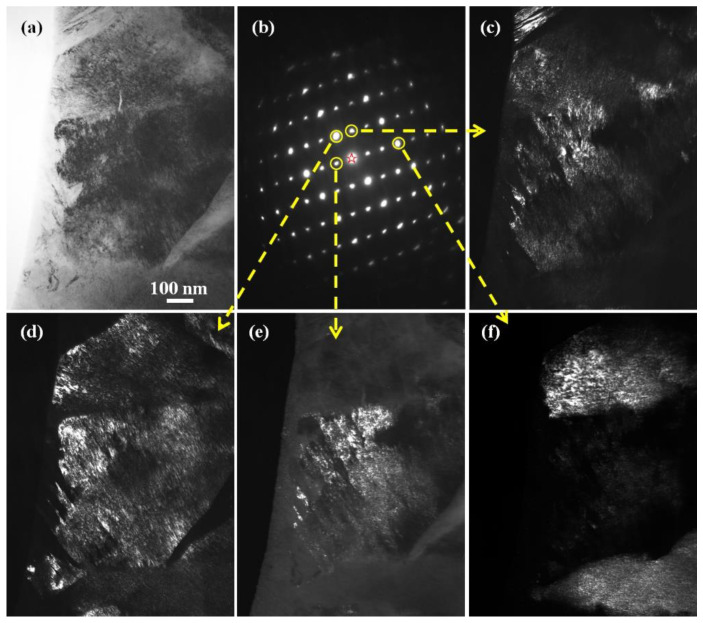
(**a**) TEM bright field image, (**b**) SAED pattern, (**c**–**f**) TEM dark field images correspond to different diffraction spots in (**b**). The electron diffraction index of (**b**) can be referred to as that in Figure 4b.

**Figure 6 materials-15-06258-f006:**
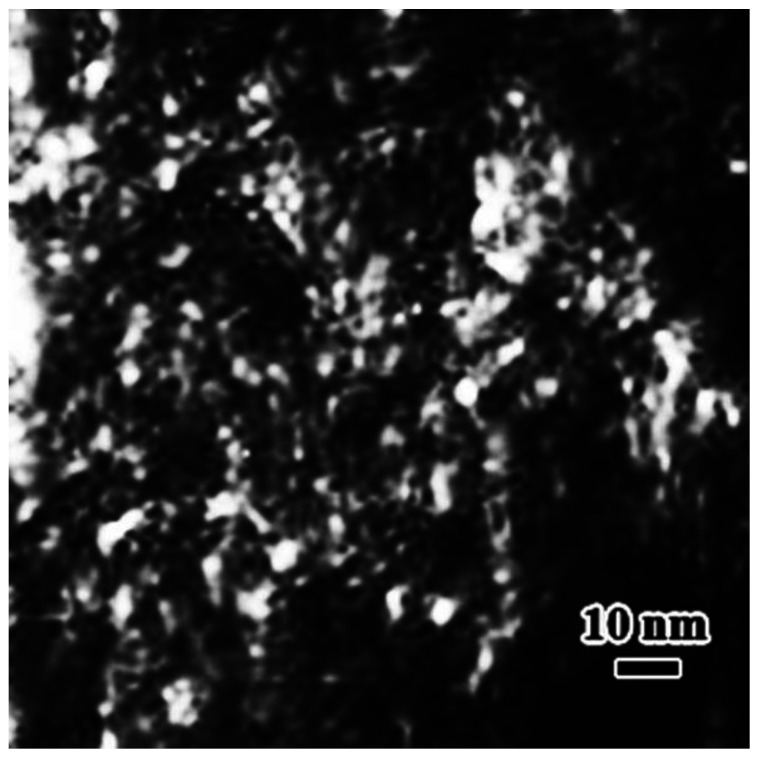
An enlarged image of a local region in Figure 5d. Several regions with a large particle size are visible. This is due to the aggregated nanocrystals in the regions having a much closer crystal orientation. It is difficult to separate them at the same contrast level. When the brightness in the regions was reduced, finer grains could be seen clearly.

**Figure 7 materials-15-06258-f007:**
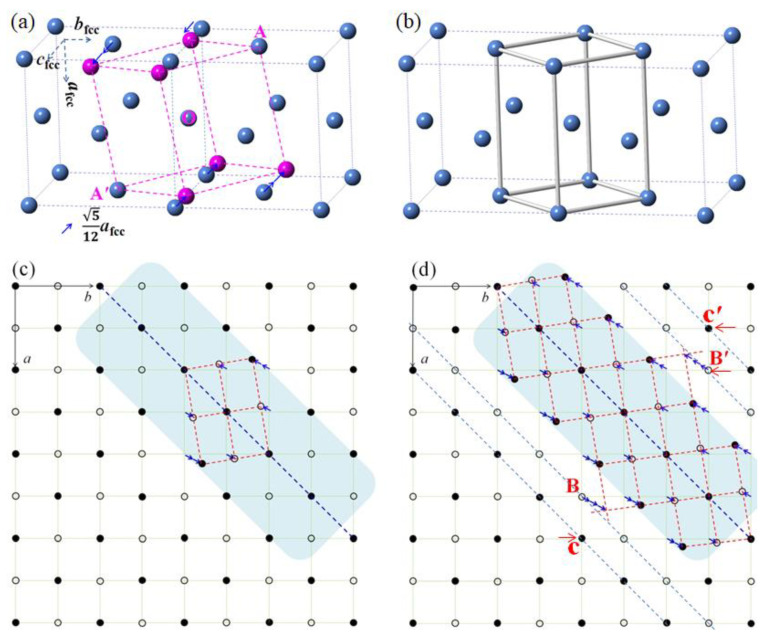
Schematic representation of the γ → α transformation process: (**a**) the model in [30]. Gray dots are the fcc-Fe atoms; the pink one is bcc-Fe, and some are not shifted at the positions of A, O, and A′. One arrow indicates that the atom shifts a distance of (√5/20)*a*_fcc-Fe_ along the arrow direction (one of the <110> directions in {100} planes). (**b**) Bain model. The atoms connected by bars are the atoms, which should transform into bcc-Fe. (**c**) The projection of (**a**) along the *c*_fcc-Fe_ direction. The atoms connected with red dashed lines are bcc-Fe atoms. The arrows are the same in (**a**,**d**). (**d**) All atoms in the shadow region transformed into bcc-Fe structure. Double or triple arrows correspond to the atoms that have shifted two or three times of one arrow distance along the arrow direction, respectively. The open circles and dots are in different layers in (**c**,**d**).

**Figure 8 materials-15-06258-f008:**
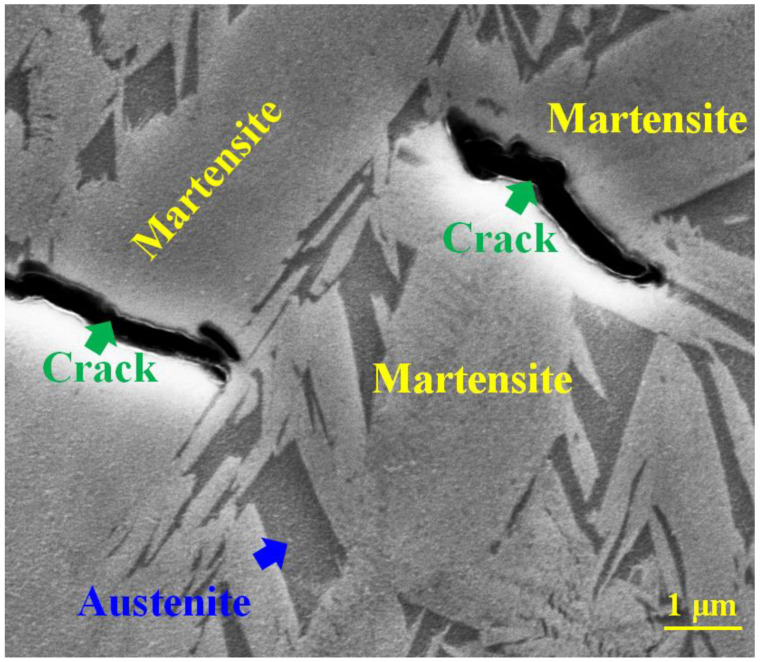
SEM image of the martensite structure in a quenched cast iron sample. The cracks with dark contrast appeared in martensite.

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
