# Peer review of "The Formation Mechanism of Nanocrystals after Martensitic Transformation"

_materials, 2022, doi:10.3390/ma15186258_

Round 1

Reviewer 1 Report

Authors investigated the martensite transformation in Fe-C steel focusing on nanocrystals formation,

The topic of the paper is of great interest and reliable results have been reported. The English of the paper is good and the article is well-structured.

I can recommend publishing the paper as it is science I couldn’t find anything minor even minor for revision.

Author Response

Response to Reviews (materials-1847882)

Dear Editors and Reviewers,

We very much appreciate the comments and suggestions from reviewer. The following is our answers to each point. Thank you very much for your consideration.

Comments and Suggestions for Authors

Authors investigated the martensite transformation in Fe-C steel focusing on nanocrystals formation, The topic of the paper is of great interest and reliable results have been reported. The English of the paper is good and the article is well-structured. I can recommend publishing the paper as it is science I couldn’t find anything minor even minor for revision.

Answer

Thank you very much, we very much appreciate the above comments.

We feel that we have answered all comments and suggestions. The English has been checked again by a native-speaker.

We would like to express our great appreciation to you and reviewer for comments on our paper. Looking forward to hearing from you.

August 12, 2022

Regards

Ping and co-authors

Reviewer 2 Report

Review of the work “The formation mechanism of nanocrystals after martensitic transformation”. The aim of this work is to obtain the formation mechanism of α-Fe phase nanocrystals after the martensitic transformation.

The authors have carried out an excellent structural characterization using different techniques. They have compared the fit of their results with different accepted theories that suggest formation mechanisms of this bcc phase from austenite. The authors have successfully predicted a mechanism that fits their experimental results.

The work is well structured and the images are of quality. In my opinion it would only take a re-reading to detect typographical errors.

Author Response

Response to Reviews (materials-1847882)

Dear Editors and Reviewers,

We very much appreciate the comments and suggestions from reviewer. The following is our answers to each point. Thank you very much for your consideration.

Comments and Suggestions for Authors

Review of the work “The formation mechanism of nanocrystals after martensitic transformation”. The aim of this work is to obtain the formation mechanism of α-Fe phase nanocrystals after the martensitic transformation.

The authors have carried out an excellent structural characterization using different techniques. They have compared the fit of their results with different accepted theories that suggest formation mechanisms of this bcc phase from austenite. The authors have successfully predicted a mechanism that fits their experimental results.

The work is well structured and the images are of quality. In my opinion it would only take a re-reading to detect typographical errors.

Answer

Thank you very much, we have re-checked the draft carefully, and typographical errors have been corrected.

We feel that we have answered all comments and suggestions. We would like to express our great appreciation to you and reviewer for comments on our paper. Looking forward to hearing from you.

August 12, 2022

Regards

Ping and co-authors

Reviewer 3 Report

The paper reports an interesting and very useful work covering mechanism of nanocrystals after martensitic transformation obserwation. The manuscript is well structured and can be published after some revisions. The reviewer enjoyed reading this paper.

The manuscript has some weaknesses. Mentioned below aspects should be taken into consideration during the revision:

1.    Intoduction:

a.    Literature analysis should be expanded. It is recommended to better justify the formation mechanism of nanocrystals and their mechanical properties influence.

2.    Experimental program:

a.    How were the mechanical properties obtained? From literature or determined by the authors for the tested material? Could the Authors present the mechanical properties in the form of an additional plot or table?

b.    Do the authors have SEM fractographies for the tested specimens? This could help to better understand the mechanisms in another scale.

3.    Conclusions:

a.    The conclusions should be in a quantified form.

b.    The practical usefulness of the results should be emphasized.

c.     The main limitations of the present method must be identified and discussed in the end of this section.

Author Response

Response to Reviews (materials-1847882)

Dear Editors and Reviewers,

We very much appreciate the comments and suggestions from reviewer. The following is our answers to each point. Thank you very much for your consideration.

Comments and Suggestions for Authors

The paper reports an interesting and very useful work covering mechanism of nanocrystals after martensitic transformation obserwation. The manuscript is well structured and can be published after some revisions. The reviewer enjoyed reading this paper.

The manuscript has some weaknesses. Mentioned below aspects should be taken into consideration during the revision:

  1. Intoduction:
  2. Literature analysis should be expanded. It is recommended to better justify the formation mechanism of nanocrystals and their mechanical properties influence.

Answer

There are numerous literatures about martensite structure, we can select those most relevant literature only. As we know that the nanocrystal alpha-Fe is first reported by us, there is no related literature, and also nobody has reported the research results on the mechanical properties related to such nanocrystal alpha-Fe in martensite, the reason is that the martensite with such ultrafine aloha-Fe grains is brittle.

  1. Experimental program:
  2. How were the mechanical properties obtained? From literature or determined by the authors for the tested material? Could the Authors present the mechanical properties in the form of an additional plot or table?

Answer

The martensite with such ultrafine aloha-Fe grains is brittle, up to now, nobody has done mechanical test.

  1. Do the authors have SEM fractographies for the tested specimens? This could help to better understand the mechanisms in another scale.

Answer

This is a good idea. We have shown a SEM image with cracks in the present draft. The detail research needs to be carried out in the future.

  1. Conclusions:
  2. The conclusions should be in a quantified form.

Answer

we shall try to quantify our results in the future draft, not in the present manuscript. This manuscript is presenting the formation mechanism of 1-2 nm alpha-Fe after martensitic transformation.

  1. The practical usefulness of the results should be emphasized.

Answer

This manuscript is presenting the formation mechanism of 1-2 nm alpha-Fe after martensitic transformation. The practical usefulness is to understand the martensitic transformation, and how to heat-treat carbon steels.

  1. The main limitations of the present method must be identified and discussed in the end of this section.

Answer

We can not fully understand the meaning of the above comment. “The present method” in our manuscript is a conventional transmission electron microscope (TEM). There is no any limitation to our present investigation.

We feel that we have answered all comments and suggestions. The English has been checked again by a native-speaker.

We would like to express our great appreciation to you and reviewer for comments on our paper. Looking forward to hearing from you.

August 12, 2022

Regards

Ping and co-authors

Reviewer 4 Report

Dear authors,

overall an interesting ivestigation.

But what is new in this publication compared to reference 21?

Apart from different materials the findings are the same. Therefore the scientific soundness is low.

Please correlate the actual findings to the findings of reference 21 and show in your discussion what is new and what are the oustanding findings. This is not clear.

Best regards

Author Response

Response to Reviews (materials-1847882)

Dear Editors and Reviewers,

We very much appreciate the comments and suggestions from reviewer. The following is our answers to each point. Thank you very much for your consideration.

Comments and Suggestions for Authors

Overall an interesting ivestigation.

But what is new in this publication compared to reference 21?

Apart from different materials the findings are the same. Therefore the scientific soundness is low.

Please correlate the actual findings to the findings of reference 21 and show in your discussion what is new and what are the oustanding findings. This is not clear.

Answer

In ref. 21, we only reported the observation results, and then we found that those ultra-fine alpha-Fe grains can quickly recrystallized into large grains. In the present manuscript, we have extended our investigation on the ultra-fine alpha-Fe grains formed just after martensitic transformation. The main difference is that we made a scientific explanation about the formation mechanism of the ultra-fine alpha-Fe grains in twinned martensite.

We feel that we have answered all comments and suggestions. The English has been checked again by a native-speaker.

We would like to express our great appreciation to you and reviewer for comments on our paper. Looking forward to hearing from you.

August 12, 2022

Regards

Ping and co-authors

Reviewer 5 Report

As per the attached review comments, the author needs to address the comments carefully.

Author Response

Response to Reviews (materials-1847882)

Dear Editors and Reviewers,

We very much appreciate the comments and suggestions from reviewer. The following is our answers to each point. Thank you very much for your consideration.

Comments and Suggestions for Authors

Question 1:

Provide the significance of Fe-C martensite.

Answer

The significance of Fe-C martensite is that such structure can cause significant hardness and strength improvement in steels, this is a well-known fact. We feel it is not necessary to mention this fact again and again.

Question 2:

Why TEM is used?

Answer

TEM is powerful in analyzing local region structure. This local region means fine to atomic level. Thus, TEM is used.

Question 3:

The present observation result matches well with the suggestion based on X-ray studies carried out one hundred years ago. More validation is needed.

Answer

Your suggestion is very good. More validation is needed! Here we provide first validation in the present manuscript. We hope other researchers can confirm or check the existence of ultra-fine alpha-Fe in quenched twinned martensite in the future. This is a basic research knowledge to the researchers in the field of metals and alloys.

Question 4:

Mention the reasons why No defect contrast can be observed inside each nanocrystal?

Answer

Each nanocrystal is just 1-2 nm size. In such fine grains, no defect can exist since any defect will cause un-stability of the fine crystal structure.

Question 5:

Write the future prospects in the abstract.

Answer

We need to carry out first-principle calculation on such ultra-fine grains, and we also need to consider magnetism on iron. We are not quite sure about the future prospects. Thus, we do not want to make some sentences to cause mis-understanding to other researchers.

Question6:  

Conclusion is well writte.

Answer

Thanks a lot.

We feel that we have answered all comments and suggestions. The English has been checked again by a native-speaker. We would like to express our great appreciation to you and reviewer for comments on our paper. Looking forward to hearing from you.

August 12, 2022

Regards

Ping and co-authors

Round 2

Reviewer 4 Report

Dear authors,

you didn't change anything from v1 to v2.

Sorry, but this is not the way of good scientific discussion!

Best regards

Author Response

Dear Reviewer,

We very much appreciate your comments. However, we do not know how to modify our manuscript since no detailed comments are listed or pointed out.

Regards

Ping

Reviewer 5 Report

The paper is now accepted and recommended for publication.

Author Response

Dear Reviewer,

We very much appreciate your comments.

Best regards

ping